# AN ENTROPIC RISK MEASURE FOR ROBUST COUNTERFACTUAL EXPLANATIONS

## ABSTRACT

Counterfactual explanations often become invalid if the underlying model changes because they are usually quite close to the decision boundary. Thus, the robustness of counterfactual explanations to potential model changes is an important desideratum. In this work, we propose entropic risk as a novel measure of robustness for counterfactual explanations. Entropic risk is a convex risk measure and satisfies several desirable properties. Furthermore, we show several ways of incorporating our proposed risk measure in the generation of robust counterfactuals. The main significance of our measure is that it establishes a connection between existing approaches for worst-case robust (min-max optimization) and robustness-constrained counterfactuals. A limiting case of our entropic-risk-based approach yields a worst-case min-max optimization scenario. On the other hand, we also provide a constrained optimization algorithm with probabilistic guarantees that can find counterfactuals, balancing our measure of robustness and the cost of the counterfactual. We study the trade-off between the cost of the counterfactuals and their validity under model changes for varying degrees of risk aversion, as determined by our risk parameter knob. We examine the performance of our algorithm on several datasets. Our proposed risk measure is rooted in large deviation theory and has close connections with mathematical finance and risk-sensitive control.

## 1 INTRODUCTION

The widespread adoption of machine learning models in critical decision-making, from education to finance (Dennis, 2018; Bogen, 2019; Chen, 2018; Hao & Stray, 2019), has raised concerns about the explainability of these models (Molnar, 2019; Lipton, 2018). To address this issue, a recently-emerging category of explanations that has gained tremendous interest is: *counterfactual explanation* (Wachter et al., 2017). In binary classification, given a specific data point and a model, a counterfactual explanation (also referred to as "counterfactual") is a feature vector leading to a different decision under the same model. Typically, counterfactuals are based on the *closest* point on the other side of the decision boundary of the model, also referred to as closest counterfactual. The closest counterfactual is a popular technique to explain a decision made by a machine learning model (see Karimi et al. (2020); Barocas et al. (2020); Mishra et al. (2021) for surveys on counterfactual explanations). For example, in automated lending models, a counterfactual can inform a denied loan applicant about specific changes, like increasing collateral, that could have led to loan approval.

However, machine learning models often undergo periodic updates, either due to more data becoming available, or due to retraining with new hyperparameters or seeds (Upadhyay et al., 2021; Black et al., 2021). Such updates can cause the closest counterfactual to become invalid by the time the user gets to act on it because these counterfactuals are typically close to the decision boundary. In our lending example, the counterfactual might indicate that a higher-valued collateral could have led to loan approval. However, if the borrower returns later with improved collateral, they could still face denial due to model updates. This not only impacts the loan applicant's trust but also the institution's reputation, and raises ethical and legal concerns. This motivates another important desideratum for counterfactuals: robustness. Robustness catalyzes the adoption of counterfactual explanations and promotes their use for high-stakes applications (Verma et al., 2020).

To make sure counterfactual explanations are useful and actionable to the users, we not only need them to be close but also require them to stay valid under model changes. In general, it might be

impossible to guarantee the existence of a counterfactual that stays valid for all possible changes to a model (see Dutta et al. (2022); Hamman et al. (2023) for impossibility results). However, one might be able to provide guarantees for a class of changes to the models. This leads to the notion of defining classes of model changes and consequently robustness of counterfactuals with respect to a specific class of model changes. The robustness of a counterfactual can then be quantified by a robustness measure, that might provide guarantees under a certain class of model changes. Then the optimization objective is to leverage the robustness measure to find a counterfactual that balances the cost of the counterfactual with respect to the original point (distance) and the robustness measure to ensure the validity of the counterfactual under model changes.

**Related Works.** The importance of robustness in local explanation methods was studied in works as early as Hancox-Li (2020), but only a handful of recent works have explored algorithms for generating robust counterfactuals. Existing literature has looked at both worst-case robust approaches, e.g. Upadhyay et al. (2021), as well as robustness-constrained approaches, e.g. Black et al. (2021) and Hamman et al. (2023). The robustness-constrained methods quantify counterfactual robustness within a Lipschitz model's local neighborhood, integrating it as a constraint in optimization to ensure the counterfactual's validity under model variations. In contrast, the worst-case robust techniques (Upadhyay et al., 2021) hedge against the worst model within a model class, employing min-max optimization to find robust counterfactuals. Notably, the connection between these two approaches has not been studied in existing literature. Other related works on the robustness of counterfactuals to model changes include Rawal et al. (2020); Dutta et al. (2022); Jiang et al. (2022); Nguyen et al. (2022). See Appendix D for a more comprehensive literature review.

We also note that there have been alternative perspectives on the robustness of counterfactuals. Two notable examples include Laugel et al. (2019); Alvarez-Melis & Jaakkola (2018) that propose an alternative perspective of robustness in explanations (called L-stability in Alvarez-Melis & Jaakkola (2018)) which is built on similar individuals receiving similar explanations, and Pawelczyk et al. (2022); Maragno et al. (2023); Dominguez-Olmedo et al. (2022) that focus on robustness to input perturbations (noisy counterfactuals) rather than model changes. In contrast to these, our focus is on counterfactuals remaining valid after model changes.

**Our Contributions.** In this work, we propose a novel entropic risk measure to quantify the robustness of counterfactuals. Entropic risk is a convex risk measure and satisfies several desirable properties. Furthermore, we show several ways of incorporating our proposed risk measure in the generation of robust counterfactuals. The main significance of our measure is its ability to establish a unifying connection between existing approaches for worst-case robust (min-max optimization) and robustness-constrained counterfactuals. Our proposed measure is rooted in large deviation theory and mathematical finance (Föllmer & Schied, 2002). Our contributions in this work are as follows:

1. **An Entropic Risk Measure for Robustness:** We propose a novel entropic risk measure to quantify the robustness of counterfactuals to model changes. Our measure is convex and satisfies several desirable properties. It has a "knob"– the risk parameter– that can be adjusted to trade-off between risk-constrained and worst-case (adversarially robust) approaches. While risk-constrained accounts for general model changes in an expected sense, the adversarial robustness prioritizes the worst-case perturbations to the model, thus having a higher cost. Our approach enables one to tune "how much" a user wants to prioritize for the worst-case model changes, by trading off cost.

2. **Connection to Min-Max Optimization:** Our proposed entropic risk measure enables us to establish the connection between the worst-case robust counterfactuals (min-max optimization P2) and the robustness-constrained counterfactuals (constrained optimization P3). The worst-case robustness approaches are in fact a limiting case of our entropic-risk-based approach (see Theorem 1). The extreme value of the knob (risk parameter) maps our measure back to a min-max/adversarial approach. By establishing this connection, we show that our proposed measure is not postulated and stems from the mathematical connection with worst-case robustness analysis.

3. **Estimation of Risk Measure with Probabilistic Guarantees:** This contribution is to serve as an example of how our theoretical contribution can be capitalized for algorithmic development. While our main contribution is a theoretical link between our method and worst-case robust counterfactuals , to the end of showcasing the potential algorithmic impact of our work, we also propose a relaxed estimator for the entropic risk measure that can be computed from sampling around the counterfactual. When the exact distribution of the changed model is known, we may be able to exactly compute our risk measure and directly solve P3, as we demonstrate through

some examples (see Examples 1 and 2). However, this may not be the case always, requiring us to explore alternate approaches for algorithmic development. In such cases, we show that under certain regularity conditions, our relaxed estimator is a lower bound to the entropic risk with high probability (see Theorem 2). This further allows us to relax optimization P3 to P4 which is amenable to implementation using Auto-differentiation (TensorFlow, 2023).

4. **Algorithm and Experimental Results:** We include an algorithm that leverages our relaxed risk measure and finds counterfactuals that are close and robust. We provide a trade-off analysis between the cost (distance) and robustness of the counterfactual for our algorithm. Our experiments are aimed at showing that our method performs on par with SNS algorithm (Black et al., 2021) while being grounded in the solid theoretical foundation of large deviation theory, without relying on any populations, enabling a unifying theory for robust counterfactuals. Our method outperforms the min-max algorithm ROAR (Upadhyay et al., 2021) across various well-known datasets.

## 2 Preliminaries

Here, we provide some contextual details, definitions, and background materials, and set our notation. We consider machine learning models $m(\cdot) : \mathbb{R}^d \to [0, 1]$ for binary classification that takes an input value $x \in \mathcal{X} \subseteq \mathbb{R}^d$ and output a probability between 0 and 1. Let $\mathcal{S} = \{x_i \in \mathcal{X}\}_{i=1}^n$ be a dataset of $n$ independent and identically distributed data points generated from an unknown density over $\mathcal{X}$.

**Definition 1** ($\gamma-$Lipschitz). *A function $m(\cdot)$ is said to be $\gamma-$Lipschitz with respect to $p$-norm if*

$$|m(x) - m(x')| \leq \gamma \|x - x'\|_p, \quad \forall x, x' \in \mathbb{R}^d$$

*where $\| \cdot \|_p$ denotes the $p$-norm.*

**Definition 2** (Closest Counterfactual $\mathcal{C}_p(x, m)$). *A closest counterfactual with respect to the model $m(\cdot)$ of a given point $x \in \mathbb{R}^d$ such that $m(x) < 0.5$ is a point $x' \in \mathbb{R}^d$ such that $m(x') \geq 0.5$ and the $p$-norm $\|x - x'\|_p$ is minimized.*

$$\mathcal{C}_p(x, m) = \arg \min_{x' \in \mathbb{R}^d} c(x, x') \quad s.t. \quad m(x') \geq 0.5.$$

For example, norm $p = 1$ results in counterfactuals with as few feature changes as possible, enforcing a sparsity constraint (also referred to as "sparse" counterfactuals (Pawelczyk et al., 2020)).

**Goals:** In this work, our goal is to obtain a systematic measure of the robustness of counterfactuals to potential model changes that satisfy desirable properties. Towards this goal, we propose an entropic risk measure that leads to a unifying connection between worst-case robustness methods (min-max optimization) and constrained-optimization-based robustness methods. Our objective involves: (i) arriving at a robustness measure for a counterfactual $x$ and a given model $m(\cdot)$, that quantifies its robustness to potential model changes; (ii) establishing the connection between our proposed-entropic-risk-based approach and the worst-case robustness approaches, and (iii) showing the algorithmic impacts of our measure by developing several algorithms for generating robust counterfactuals based on our robustness measure. The existing methods for finding robust counterfactuals can be divided into either worst-case robust or robustness-constrained approaches. Our research closes the gap between these two perspectives by showing that the worst-case approach is a limiting case of our proposed entropic risk-based approach. Furthermore, we also propose a relaxed estimator for our risk measure that (i) is an upper-bound to our entropic risk measure with high probability; and (ii) can be computed easily by sampling around the counterfactual and incorporated into the optimization.

## 3 Main Results: Robustness via Entropic Risk Measure

Robustness is essential for a counterfactual to be trusted as a reliable explanation of the model's predictions. The robustness is achieved at the expense of a higher cost, resulting in a counterfactual that is further away from the original input vector. This means that to find a robust counterfactual with respect to a changed model, we need to balance the cost and robustness of the counterfactual. To ensure the counterfactual is valid under all plausible model changes, we formulate a general multi-objective optimization that hedges against the worst model change and balances the cost and robustness of the worst model, i.e.,

$$\min_{x' \in \mathbb{R}^d} (c(x, x'), \max_{M \in \mathcal{M}} \ell(M(x'))) \quad \text{s.t.} \quad m(x') \geq 0.5. \tag{P}$$

Here $c : \mathcal{X} \times \mathcal{X} \to \mathbb{R}_+$ is the cost of changing an instance $x$ to $x'$, e.g., $c(x, x') = \|x - x'\|_p$ where $1 \leq p \leq \infty$, and $\ell : \mathcal{M} \times \mathcal{X} \to R_+$ is a differentiable loss function which ensures that $M(x')$ is close to the desired value of 1, e.g., $\ell(M(x)) = 1 - M(x)$. We denote the changed model by $M(\cdot) : \mathbb{R}^d \to [0, 1]$. Recall that $m(\cdot)$ denotes a given model and is not random. The second objective function $\max_{M \in \mathcal{M}} \ell(M(x'))$ is the worst-case loss over all possible model changes in the set $\mathcal{M}$.

To address a multi-objective optimization problem of this nature, we can seek the Pareto optimal front using established techniques, such as linear scalarization or the epsilon-constraint methods (Miettinen, 1999). The linear scalarization approach, for instance, entails solving

$$\min_{x' \in \mathbb{R}^d} \max_{M \in \mathcal{M}} c(x, x') + \lambda \ell(M(x')) \quad \text{s.t.} \quad m(x') \geq 0.5 \tag{P1}$$

for different values of $\lambda$ to generate Pareto optimal solutions (e.g., a relaxed variant of this approach is employed in Upadhyay et al. (2021)), meanwhile, the epsilon-constraint method addresses the problem by solving

$$\min_{x' \in \mathbb{R}^d} c(x, x') \quad \text{s.t.} \quad \max_{M \in \mathcal{M}} \ell(M(x')) < \tau, \quad m(x') \geq 0.5 \tag{P2}$$

for different values of $\tau$ (e.g., a relaxed variant of this approach is employed in Hamman et al. (2023)).

By varying $\lambda$ in P1 or $\tau$ in P2, different points on the Pareto front can be obtained (also see the book Miettinen (1999)). To see the equivalence of the threshold $\tau$ and the multiplier $\lambda$, note that the sensitivities of the cost $c(x, x')$ with respect to changes in the threshold $\tau$ (evaluated at the optimal $x'^*$) is the negative of the optimal multiplier (dual variable) $\lambda$ (for a background on multi-objective optimization, please refer to Appendix A.4 (Castillo et al., 2008)), i.e, $\partial c(x, x'^*)/\partial \tau = -\lambda^*$. Each $\lambda$ and $\tau$ results in a point on the Pareto optimal front of the multi-objective optimization problem (Miettinen, 1999; Castillo et al., 2008). Both P1 and P2 lead to the same Pareto front, and $\lambda$ and $\tau$ can be chosen such that P1 and P2 have the same solutions. The Pareto front characterizes the trade-off between the cost and robustness of the counterfactuals.

The worst-case loss $\max_{M \in \mathcal{M}} \ell(M(x'))$ hedges against the worst possible model, but can often lead to somewhat conservative counterfactuals, i.e., ones which are quite well within the boundary and have a high cost (distance). To mitigate this issue, we use a risk measure that allows us to *hedge against the models based on their probability of occurrence*. We assume the changed model $M$ is drawn from a probability distribution $P$ over the set of models $\mathcal{M}$. Here, we propose the entropic risk measure as a quantification of robustness for counterfactuals which is defined as follows:

**Definition 3.** *The entropic risk measure of a random variable with the risk aversion parameter $\theta > 0$ is denoted by $\rho_\theta^{ent}(\cdot)$ and is given by:*

$$\rho_\theta^{ent}(\ell(M(x'))) := \frac{1}{\theta} \log(\mathbb{E}_{M \sim P}[e^{\theta \ell(M(x'))}]), \quad \theta > 0. \tag{1}$$

The parameter $\theta$ is called the risk parameter. A positive risk parameter results in risk-averse behavior. Hence, we refer to a positive risk parameter as the risk-aversion parameter. We show in Theorem 1 that as we increase the risk-aversion parameter, our probabilistic method converges to a worst-case formulation. Definition 3 allows us to reformulate our problem as follows:

$$\min_{x' \in \mathbb{R}^d} c(x, x') \quad \text{s.t.} \quad \rho_\theta^{ent}(\ell(M(x'))) < \tau, \quad m(x') \geq 0.5. \tag{P3}$$

**Properties of Entropic Risk Measure.** Entropic risk measure is rooted in large deviation theory and is not postulated. This measure enables establishing a connection to worst-case approaches for finding counterfactuals. Taylor's expansion of the exponential shows that the entropic risk measure is the infinite sum of the moments of the distribution. Furthermore, it is well-known (Föllmer & Schied, 2002) that entropic risk measure is a *convex* risk measure and as such, for a positive risk parameter $\theta > 0$, satisfies the properties of (1) monotonicity, (2) translation-invariance, and (3) convexity.

1. **Monotonicity.** $\ell(M_1(\cdot)) \geq \ell(M_2(\cdot)) \Rightarrow \rho_\theta^{ent}(\ell(M_1(\cdot))) \geq \rho_\theta^{ent}(\ell(M_2(\cdot)))$.

2. **Translation invariance.** For constant $\alpha \in \mathbb{R}$, $\rho_\theta^{ent}(\ell(M(\cdot)) + \alpha) = \rho_\theta^{ent}(\ell(M(\cdot))) + \alpha$.

3. **Convexity.** For $\alpha \in [0, 1]$,

$$\rho_\theta^{ent}(\alpha \ell(M_1(\cdot)) + (1 - \alpha)\ell(M_2(\cdot))) \leq \alpha \rho_\theta^{ent}(\ell(M_1(\cdot))) + (1 - \alpha)\rho_\theta^{ent}(\ell(M_2(\cdot))).$$

For the sake of simplicity, consider the choice of cost function $\ell(M(x)) = 1 - M(x)$. Then, the monotonicity implies that a model with greater output probabilities has less risk. The translation invariance implies that adding a constant to the output of the predictor effectively reduces the risk by the same amount. The convexity is quite desirable since it means that the risk for a combined model is lower than the risk for the two of them individually.

To gain a deeper understanding of the risk constraint described in P3, we examine distributions characterized by their analytical Moment Generating Functions (MGFs). Two notable examples of such distributions are the Uniform and truncated Gaussian distributions. For simplicity, we use the cost function $\ell(M(x'))=1-M(x')$. In our formulation, this loss function is minimized, encouraging a counterfactual with a higher predicted value. When using this specific cost function, any value of the threshold $\tau$ outside the interval $[0, 1]$ renders the problem infeasible. Given these choices for the cost and model distribution, we provide the explicit form of the constraint in P3.

**Example 1.** *Let the distribution of the output of the changed model at the counterfactual point, $M(x')$, follow a uniform distribution on a $\delta$-ball around the output of the original model $m(x')$, i.e., $M(x') \sim \mathcal{U}[m(x') - \delta, m(x') + \delta]$ for some $\delta > 0$. With these choices, the constraint in P3 becomes:*

$$m(x') > (1 - \tau) + K_{\delta,\theta}, \quad K_{\delta,\theta} := \frac{1}{\theta} \log(\frac{e^{\theta\delta} - e^{-\theta\delta}}{2\theta\delta}).$$

For the Uniform distribution, due to the monotonicity of $K_{\delta,\theta}$ with respect to $\theta$, as the value of $\theta$ increases, a higher value of $m(x')$ is required to satisfy the constraint. It can be verified that $K_{\delta,\theta}$ in limit of $\theta \to \infty$ is $\delta$. Given this, for the case when $\theta \to \infty$, our constraint becomes $m(x') > 1-\tau+\delta$. As the value of $\theta$ approaches to 0, $K_{\delta,\theta}$ approaches 0 and the constraint becomes $m(x') > (1 - \tau)$, i.e., finding counterfactual $x'$ with just high $m(x')$.

**Example 2** (Truncated Gaussian). *Let the distribution of the output of the changed model at the counterfactual point, $M(x')$, follow a truncated Gaussian distribution with a mean equal to the output of the original model $m(x')$ and a variance of $\sigma^2$ that lies between 0 and 1. With these choices, the constraint in P3 becomes:*

$$m(x') > (1 - \tau) + \theta\frac{\sigma^2}{2} + \frac{1}{\theta} \log(K_\theta), \quad K_\theta := \frac{\Phi(\beta + \sigma\theta) - \Phi(\alpha + \sigma\theta)}{\Phi(\beta) - \Phi(\alpha)}$$

*where $\alpha := \frac{-\mu}{\sigma}$ and $\beta := \frac{1-\mu}{\sigma}$ and $\Phi(x) = 1/2(1 + \mathrm{erf}(x/\sqrt{2}))$. The error function, denoted by $\mathrm{erf}$, is defined as $\mathrm{erf}\, z = 2/\sqrt{\pi} \int_0^z e^{-t^2}\, \mathrm{d}t$.*

As the $\theta$ approaches 0, our constraint becomes $m(x') > 1 - \tau$. As the value of $\theta$ increases, greater weight is placed on the variance term, emphasizing its importance. In both examples, when the distributions are unknown, determining the precise threshold for model output to satisfy the constraint becomes challenging. This is because higher values are more conservative (less risky), but incur higher costs. To address this challenge, we must devise techniques that do not rely the explicit knowledge of the distribution, as explored further in the next subsections.

## 3.1 Connection of Entropic-Risk-Based Approach with Worst-Case Robustness

The two main approaches to finding robust counterfactuals: (i) with hard guarantees by hedging against the worst-case; and (ii) with probabilistic guarantees by leveraging robustness constraints can be bridged by leveraging our entropic risk measure. We first establish the connection between our risk-based and the worst-case formulations in the following theorem. The theorem states that the worst-case approach is the limiting case of our risk-based method as $\theta \to \infty$.

**Theorem 1.** *In the limit as the risk-aversion parameter $\theta$ approaches infinity, the optimization P3, which involves constraining the entropic risk measure associated with the robustness of models within a specific model class, asymptotically converges to the optimization problem P2, where the constraint pertains to the robustness of the worst model within the same model class.*

Theorem 1 shows how the entropic risk measure provides a single parameter (knob) that determines the risk-aversion of the counterfactual and can be used to study the effect of risk-aversion on the behavior of algorithms that generate robust counterfactuals.

**Proof:** We discuss the proof in Appendix A. The proof uses Vardhan's Lemma presented here.

**Lemma 1.** *(Föllmer & Schied, 2002) Let $X$ be a random variable. The entropic risk measure is a convex risk measure and as such has a dual representation. The dual representation with the risk aversion parameter $\theta > 0$ is given by*

$$\rho_\theta^{\text{ent}}(X) = \frac{1}{\theta} \log \left( \mathbb{E}_{X \sim P}[e^{\theta X}] \right) = \sup_{Q \ll P} \left\{ E_Q[X] - \frac{1}{\theta} D(Q|P) \right\}$$

*where $D(Q|P) := \mathbb{E}_Q \left[ \log \, dQ/dP \right]$ is the Kullback-Libeler (KL) divergence between distributions $P$ and $Q$, and $Q \ll P$ denotes the distribution $Q$ is absolutely continuous with respect to $P$.*

## 3.2 Towards Estimation of The Risk Measure with Probabilistic Guarantees

We previously showed (see Examples 1 and 2) that when distributions are known, we can often compute and directly solve P3. For other cases, when the distributions are not known or are complex and the entropic risk measure is not computable due to the expectation over the changed models, we now use a high probability upper-bound on the entropic risk measure that can be computed by sampling the input space around the counterfactual (see Theorem 2). We use this upper-bound to propose a computable risk measure to quantify the robustness of counterfactuals.

**Proposed Relaxation.** We first introduce the following computable risk measure and discuss its properties. We then discuss its merit and its connections with entropic risk measures.

**Definition 4.** *The relaxed entropic risk measure for the robustness of a counterfactual $x$ is given by: $\frac{1}{k} \sum_{i=1}^k e^{\theta \ell(m(X_i))}$, where the $k$ data points are drawn from a Gaussian distribution $\mathcal{N}(x, \sigma^2 \mathrm{I}_d)$ where $\mathrm{I}_d$ is the identity matrix. The constant $\theta > 0$ is a design parameter.*

This relaxed risk measure is computable by using the evaluation of the model in points sampled around a candidate counterfactual. Thus, we are able to reformulate our optimization as follows:

$$\min_{x' \in \mathbb{R}^d} c(x, x') \quad \text{s.t.} \quad (1/\theta) \log \frac{1}{k} \sum_{i=1}^k (e^{\theta \ell(m(X_i))}) < \tau, \quad m(x') \geq 0.5. \tag{P4}$$

Here the $k$ samples are drawn from a normal distribution $N(x', \sigma^2 I)$.

For concreteness and simplicity, we focus on the loss function $\ell(M(x)) := 1 - M(x)$. The choice of $\ell$ is not crucial and all our results hold for any differentiable $\ell$. Then we have:

$$\min_{x' \in \mathbb{R}^d} c(x, x') \quad \text{s.t.} \quad R_{\theta,k}(x', m) := (1/\theta) \log \frac{1}{k} \sum_{i=1}^k (e^{-\theta m(X_i)}) < \tau - 1, \quad m(x') \geq 0.5. \tag{P5}$$

This estimator of entropic risk, defined by $R_{\theta,k}(x', m) := (1/\theta) \log \frac{1}{k} \sum_{i=1}^k (e^{-\theta m(X_i)})$, is amenable to implementation using automatic differentiation (TensorFlow, 2023) as discussed in Section 4.

**Properties of the Proposed Measure:** Before discussing our implementation of P5, we also briefly discuss some desirable properties of our relaxed risk measure here.

**Proposition 1.** *Our proposed robustness measure lies between $[-1, 0]$, that is*

$$-1 \leq R_{\theta,k}(x', m) \leq 0, \quad \theta > 0$$

For a given risk parameter $\theta$, the measure $R_{\theta,k}(x', m) = -1$ when the model outputs $m(x) = 1$ for all $x$ and $R_{\theta,k}(x', m) = 0$ when it outputs $m(x) = 0$.

**Remark 1.** *$R_{\theta,k}(x', m)$ is a scaled $LogSumExp$ function. $LogSumExp$ is a smooth approximation to the maximum function which becomes more exact as the scaling parameter $\theta$ increases.*

**Proposition 2.** *This metric also inherits the entropic risk measure properties of (1) monotonicity, (2) translation-invariance, and (3) convexity. The proofs are included in Appendix A.*

**Remark 2.** *Our proposed measure aligns with the desired properties of a robustness measure discussed in Dutta et al. (2022): The counterfactual is less likely to be invalidated under model changes if the points around the counterfactual have high model outputs and low variability.*

**The Merit of the Proposed Relaxed Measure.** When distributions of changed models are known, our risk measure (see Definition 3) can be computed and P3 is directly solvable (see Examples 1 and 2). When the distributions are not known, our measure is not computable since it needs computing expectations over the changed models. However, in cases where our setup permits us to assume that both the distributions of $m(X_i)$ and $M(x')$ characteristics and the models exhibit Lipschitz continuity, we can use this to employ an alternative, computable metric. This metric samples around the counterfactual and uses the original model outputs at those points.

Here, we assume the Lipschitness of the models and that the MGFs of the changed models at the counterfactual ($M(x')$) and the output of the original model at points chosen randomly around the counterfactual ($m(X_i)$) are relatively close. Within this framework, we deliberately opt for relatively strong assumptions on MGFs, primarily for the purpose of illustration, showcasing the potential for algorithmic advancements using our theoretical insights. Relaxing this assumption, while requiring more intricate proofs, opens the door to less assertive yet still valuable probabilistic guarantees.

MGF characterizes the properties of a random variable and can uniquely identify the distribution of a random variable by a certain set of moments. The MGF of a random variable $X$ is defined as $\mathbb{E}[e^{\theta X}]$, where '$\theta$' is a real-valued parameter. We introduce a class of model changes based on their MGF and derive probabilistic guarantees for our proposed robustness metric under such class of model changes.

**Definition 5** (MGF $(\tilde{\epsilon}, \theta)$-Equivalence). *The MGF $(\tilde{\epsilon}, \theta)$-Equivalence class of model changes for a given model $m$ is the set of all models $\mathcal{M}$ such that the following hold:*

$$\left| \mathbb{E}\left[ e^{\theta \ell(M(x))} - \frac{1}{k} \sum_{i=1}^{k} e^{\theta \ell(m(X_i))} \right] \right| < \tilde{\epsilon}$$

*where $M$ and $X_i \sim N(x, \sigma^2 I)$ are random variables.*

This condition ensures that all moments of the distribution of $M(x')$ at the counterfactual point would stay close to the average behavior of the original model $m$ within the neighborhood of the counterfactual $x'$. This can be seen by the Taylor expansion of the exponential function which is the infinite sum of its moments.

**Assumption 1** (Liptschitz Countinuity). *We assume the original model $\ell(m(x))$ is Lipschitz continuous with respect to $x$ with constant $\gamma$.*

Assumption 1 on Lipschitz continuity of the model is a critical factor enabling our sampling approach to offer a computable metric for counterfactuals. This continuity ensures that the sampled points around a counterfactual remain meaningfully related to the models' output. Since the neural networks can often satisfy local Lipschitz continuity (Virmaux & Scaman, 2018), this assumption does not impose significant restrictions on the feasibility of our method. However, a limitation of our work is that a large Lipschitz constant can weaken the robustness guarantees. Now, we formally state the probabilistic guarantee for our method.

**Theorem 2** (Probabilistic Guarantees). *Let $(X_1, X_2, \ldots, X_k)$ be points drawn from a Gaussian distribution centered around a point $x$, $\mathcal{N}(x, \sigma^2 I_d)$ where $I_d$ is the identity matrix. Let $M$ be a $(\tilde{\epsilon}, \theta)$-Equivalent model for the model $m$. Then under Assumption 1, for all $\epsilon > 2\tilde{\epsilon}$, the following upper-bound holds*

$$\Pr\left( \mathbb{E}[e^{\theta \ell(M(x))}] \leq \frac{1}{k} \sum_{i=1}^{k} e^{\theta \ell(m(X_i))} + (\epsilon + \tilde{\epsilon}) \right) > 1 - \exp\left( -\frac{k\epsilon^2}{2\gamma_e^2 \sigma^2} \right) \text{ for all } \epsilon, \tilde{\epsilon} > 0.$$

*where $\gamma_e := \gamma \theta e^{\theta}$. The constant $\theta > 0$ is a design parameter that is bounded.*

**Proof Sketch:** The proof is in the appendix. It uses the well-known concentration bound for Lipschitz functions of Gaussians (Boucheron et al., 2013).

## 4 EXPERIMENTAL RESULTS

Our main contribution in this paper is to demonstrate a theoretical connection between our method and the worst-case robust counterfactuals. Our experiments here simply aim to showcase that our

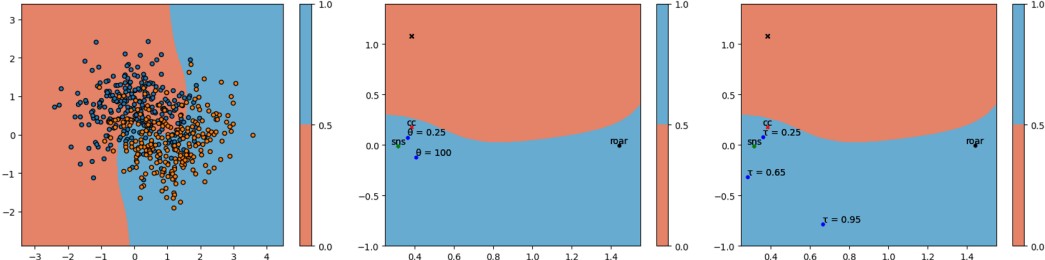

Figure 1: The left figure plots the data points and the decision boundary of the trained model for the Moon dataset. The middle figure plots a randomly chosen data point and it's closest counterfactual (cc) alongside its robust counterfactuals obtained by ROAR (a worst-case min-max method), SNS (a risk-constrained method), and our method for a given threshold $\tau$=0.25 and a varying risk parameter $\theta$={0.25, 100}. The right figure plots the same randomly chosen data point and its closest counterfactual (cc) alongside its robust counterfactuals obtained by ROAR, SNS, and our method for a given risk parameter $\theta$=0.25 and a varying threshold $\tau$={0.25, 0.65, 0.95}.

method performs on par with SNS (Black et al., 2021) algorithm, while being grounded in the solid theoretical foundation of large deviation theory without relying on any postulations, enabling a unifying theory for robust counterfactuals. Furthermore, our method outperformed the min-max type algorithm ROAR (Upadhyay et al., 2021).

**Algorithmic Strategy.** To solve P5, we use a counterfactual generating algorithm and optimize our risk constraint iteratively. To incorporate our method, we compute the robustness metric for the generated counterfactual and, if needed, a gradient ascent process updates the counterfactual until a robust counterfactual that meets the desired robustness is found. This process can leverage any existing method to generate counterfactual $x'$ for $x$. We check if $R_{k,\sigma^2}(x', m) \leq \tau - 1$ which in that case outputs the counterfactual $x'$. Otherwise, it performs a gradient descent step $x' = x' - \eta \Delta_{x'} R_{k,\sigma^2}(x', m)$ in the direction of the gradient of the robustness measure $\Delta_{x'} R_{k,\sigma^2}(x', m)$. This step repeats until a suitable $x'$ is found or a maximum number of steps is reached. The differentiation is performed using Auto-differentiation (TensorFlow, 2023). This is similar to T-Rex algorithm proposed in Hamman et al. (2023) which we use in conjugation with our proposed risk measure. We refer to our method as Entropic T-Rex.

**(Synthetic) 2D Dataset.** To enhance the clarity and comprehension of our ideas, we first present experimental results using a synthetic 2D dataset. The 2D dataset allows visual demonstration to showcase the core principles of our proposed algorithm and the way it is bridging the gap between the two approaches in robust counterfactual. We generated 500 sample data points from the synthetic moon dataset of *Scikit-learn* with the noise parameter set to 0.55. We trained a neural network with 3 hidden layers each with 128 neurons and tanh activation function. Figure 1 (left) shows the data points and the decision boundary of the trained model. In Figure 1 (middle), a randomly selected data point and its closest counterfactual are depicted, along with its robust counterfactuals obtained through three distinct methods: ROAR (a worst-case min-max approach), SNS (a risk-constrained method), and our proposed method. This shows that the counterfactual point progressively keeps moving inwards to avert more risk as $\theta$ is increased. This illustration is presented with a fixed threshold and varying risk parameters. Similarly, in Figure 1 (right), the same randomly selected data point, its closest counterfactual, and robust counterfactuals are showcased using ROAR, SNS, and our method, but this time with a fixed risk parameter and varying threshold values.

**Benchmark Datasets.** We present our experimental results on existing datasets. Our experimental results confirm our theoretical understanding and show the efficacy of our proposed method.

**Datasets.** We run our experiments on a number of benchmark datasets. The results for the German Credit (Dua & Graff, 2017) and HELOC (FICO, 2018) dataset are reported here. We report the results for the HELOC dataset here. The results for the other datasets are qualitatively similar and are included in Appendix B. The results for the dataset Cardiotocography (CTG) (Dua & Graff, 2017) are in Appendix B. We normalize the feature vectors to lie between [0, 1] for simplicity.

**Performance Metrics.** Robust counterfactual-generating algorithms balance the cost (distance) and robustness (validity). In particular, we consider: (i) **Cost:** average $l_1$ or $l_2$ distance between counterfactuals $x'$ and the input point $x$ and (ii) **Validity(%):** The percentage of counterfactuals that remain valid under the new model $M$.

**Methodology.** We train 51 neural networks to find counterfactuals. We use one neural network, $m$, to find the counterfactuals and the other 50, $M$, to evaluate the robustness of the counterfactual. To evaluate the $validity$ of counterfactuals, the model changes include (i) Training models using the same hyperparameters but different weight initialization; and (ii) Retraining models by randomly removing a small portion, $1\%$, of the training data each time and using different weight initialization.

**Hyperparameter Selection.** We choose $k=1000$. This was sufficient to ensure a high probability bound, but to keep the computational requirements of our algorithm in check. We chose $\sigma$ by analyzing the standard deviation of the features. We experimented with some hyperparameters and selected a value of $\sigma = 0.1$.
The parameters $\tau'(=1-\tau)$ and $\theta$ are specific to our proposed robustness measure. The threshold $\theta$ is a hyper-parameter that trade-offs the cost for the desired effective validity. Recall $\theta$ is the risk-aversion parameter and in the limit as $\theta \to \infty$ our proposed measure becomes a worst-case measure. Higher values of $\tau'$ result in counterfactuals with higher validity (robustness), and higher cost. There is an interaction between $\tau'$ and $\theta$, so these parameters need to be chosen together. Here we report the results for $\theta = 1$ and $\tau' = 0.2$.

Table 1: Experimental results for HELOC dataset.

| **HELOC** | $l_1$ based | | $l_2$ based | |
|---|---|---|---|---|
| | COST | VAL. | COST | VAL. |
| Closest Counterfactual | 0.45 | 69% | 0.54 | 77% |
| **Entropic T-Rex** (ours) | 2.82 | 100% | 0.76 | 100% |
| SNS | 1.25 | 98% | 0.33 | 98% |
| ROAR | 3.14 | 100% | 1.22 | 100% |

**Results.** Our experimental results are consistent with our theoretical analysis. We observe the minimum cost counterfactual may not be robust. Our proposed method, Entropic T-Rex, provides a knob that can be adjusted to control and improve the validity of the counterfactuals, at the price of higher cost. Our algorithm achieves comparable results to the SNS method and T-Rex. The advantage of our algorithm is in the theoretical results that connect it to the robust worst-case model. The averages are reported in the table and we report the standard deviation in Appendix B.

**Trade-Off Analysis.** Figure 2 shows the trade-off between the cost and robustness. To study the behavior of our algorithm and understand the effects of the threshold $\tau$ on the cost and robustness of the counterfactuals. We run experiments and keep all hyper-parameters the same except the threshold $\tau'$.

## 5 CONCLUSION & LIMITATIONS

With our entropic risk measure, we showed that the risk-aversion parameter can be adjusted for balancing cost and robustness of counterfactuals by considering the impact of the worst model. We showed that the worst-case approach is a limiting case of our approach based on entropic risk measures. This establishes the connection between our approach and the worst-case approaches and explains the robust nature of the counterfactuals generated by our algorithm. Though

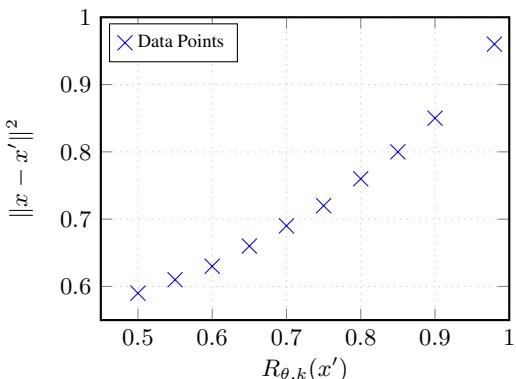

Figure 2: $\|x - x'\|^2$ vs. $R_{\theta,k}(x')$ on HELOC.

our method is practically implementable and potentially applicable to all model changes and datasets, our guarantees may not apply to all models or datasets due to our assumptions. Another limitation of our work is the dependence of our guarantee on the Lipschitz constant of the models since a large constant can weaken the guarantees. The reliance of our method on sampling and gradient estimation has the drawback of having high computational complexity. A more comprehensive discussion is included in Appendix C.

## 6 REPRODUCIBILITY STATEMENT

For the synthetic dataset, the point is chosen randomly. For the real datasets, we repeated the experiments and have reported the standard deviation alongside the average performance within the tables featured in our Appendix. The code will be release on acceptance.

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

# A  BACKGROUND AND PROOFS

## A.1  PROOF OF THEOREM 1

**Theorem 1.** *In the limit as the risk-aversion parameter $\theta$ approaches infinity, the optimization P3, which involves constraining the entropic risk measure associated with the robustness of models within a specific model class, asymptotically converges to the optimization problem P2, where the constraint pertains to the robustness of the worst model within the same model class.*

Theorem 1 shows how the entropic risk measure provides a single parameter (knob) that determines the risk-aversion of the counterfactual and can be used to study the effect of risk-aversion on the behavior of algorithms that generate robust counterfactuals.

The proof of Theorem 1 uses the results in Lemma 1 and 2.

**Lemma 1.** *(Föllmer & Schied, 2002) Let $X$ be a random variable. The entropic risk measure is a convex risk measure and as such has a dual representation. The dual representation with the risk aversion parameter $\theta > 0$ is given by*

$$\rho_\theta^{\text{ent}}(X) = \frac{1}{\theta} \log \left( \mathbb{E}_{X \sim P}[e^{\theta X}] \right) = \sup_{Q \ll P} \left\{ E_Q[X] - \frac{1}{\theta} D(Q|P) \right\}$$

*where $D(Q|P) := \mathbb{E}_Q \left[ \log \, {}^{dQ}/_{dP} \right]$ is the Kullback-Libeler (KL) divergence between distributions $P$ and $Q$, and $Q \ll P$ denotes the distribution $Q$ is absolutely continuous with respect to $P$.*

Note that $Q$ is absolutely continuous with respect to $P$ if $Q(x) = 0$ when $P(x) = 0$. This assumption ensures that the KL divergence is finite. Then, we have,

$$\lim_{\theta \to \infty} \rho^{\text{ent}}(X) = \sup_{Q \ll P} \left\{ E_Q[X] \right\}. \tag{2}$$

For simplicity, we let both $Q(\tilde{m}) > 0$ and $P(\tilde{m}) > 0$ over the set of models $\mathcal{M}$ which is a compact and bounded set. Next, we show the following result.

**Lemma 2.** *Let $Q$ be any probability distribution over the set of models $\mathcal{M}$ such that $Q(\tilde{m}) > 0$ everywhere, and $\mathcal{M}$ be a compact and bounded set. Then we have,*

$$\sup_Q \mathbb{E}_Q[\ell(M)] = \max_{m_i \in \mathcal{M}} \ell(m_i)$$

*Proof.* We prove the equality by establishing two directions of the inequality. First, we note that the expected value of a set of values is always less than or equal to its maximum value. Thus,

$$\mathbb{E}_Q[\ell(M)] \leq \max_{m \in \mathcal{M}} \ell(m), \quad \forall Q$$

Since it holds for all $Q$'s we have

$$\sup_Q \mathbb{E}_Q[\ell(M)] \leq \max_{m \in \mathcal{M}} \ell(m) \tag{3}$$

To prove the reverse direction, let $Q_m$ be a probability distribution such that

$$Q_m(\tilde{m}) = \begin{cases} 1 - \delta & \tilde{m} = m \\ \delta_{\tilde{m}} & \tilde{m} \neq m \end{cases}$$

where $\delta_{\tilde{m}} \neq 0$, for all $\tilde{m} \in \mathcal{M}$ and $\delta = \sum_{\tilde{m} \in \mathcal{M}, \tilde{m} \neq m} \delta_{\tilde{m}}$. Then, we have

$$\mathbb{E}_{Q_m}[\ell(M)] = (1 - \delta)\ell(m) + \sum_{\tilde{m} \in \mathcal{M}, \tilde{m} \neq m} \delta_{\tilde{m}} \ell(\tilde{m}), \quad \forall m$$

Thus,

$$\sup_Q \mathbb{E}[\ell(M)] \geq E_{Q_m}[\ell(M)] = (1 - \delta)\ell(m) + \sum_{\tilde{m} \in \mathcal{M}, \tilde{m} \neq m} \delta_{\tilde{m}} \ell(\tilde{m}), \quad \forall m$$

Let $m^* = \arg\max_m \ell(m)$. Then we have,

$$\sup_Q \mathbb{E}[\ell(M)] \geq (1-\delta)\ell(m^*) + \sum_{\tilde{m} \in \mathcal{M}, \tilde{m} \neq m^*} \delta_{\tilde{m}} \ell(\tilde{m})$$

By noting that $\delta$ can be made arbitrarily small, we have

$$\sup_Q \mathbb{E}[\ell(M)] \geq \max_{m \in \mathcal{M}} \ell(m) - \epsilon(\delta)$$

for an arbitrarily small $\epsilon(\delta) > 0$. Thus the result holds.

The set $\mathcal{M}$ needs to be such that the maximum exists, e.g., a bounded and compact set. $\qquad\square$

Now using Lemma 2, we have

$$\lim_{\theta \to \infty} \rho_\theta^{ent}(\ell(m(x'))) := \frac{1}{\theta} \log(\mathbb{E}_{M \sim P}[e^{\theta\ell(M(x))}]) \overset{(a)}{=} \sup_{Q \in \mathcal{M}_1} \{E_Q[\ell(M(x))]\} \overset{(b)}{=} \sup_{m \in \mathcal{M}} \ell(m(x')),$$

where (a) holds since $\lim_{\theta \to \infty} \rho^{ent}(X) = \max_{Q \ll P} \{E_Q[X]\}$ as shown in equation 2 and (b) follows from Lemma 2.

### A.2 PROOF OF THEOREM 2

**Theorem 2** (Probabilistic Guarantees). *Let $(X_1, X_2, \ldots, X_k)$ be points drawn from a Gaussian distribution centered around a point $x$, $\mathcal{N}(x, \sigma^2 \mathrm{I}_d)$ where $\mathrm{I}_d$ is the identity matrix. Let $M$ be a $(\tilde{\epsilon}, \theta)$-Equivalent model for the model $m$. Then under Assumption 1, for all $\epsilon > 2\tilde{\epsilon}$, the following upper-bound holds*

$$\Pr\left(\mathbb{E}[e^{\theta\ell(M(x))}] \leq \frac{1}{k}\sum_{i=1}^k e^{\theta\ell(m(X_i))} + (\epsilon + \tilde{\epsilon})\right) > 1 - \exp\left(-\frac{k\epsilon^2}{2\gamma_e^2\sigma^2}\right) \text{ for all } \epsilon, \tilde{\epsilon} > 0.$$

*where $\gamma_e := \gamma\theta e^\theta$. The constant $\theta > 0$ is a design parameter that is bounded.*

The proof relies on the Lipschitz property of $\ell(M)$ and its boundedness. The proof uses the results in the following lemmas.

**Lemma 3** (Gaussian Concentration Inequality). *Let $X = (X_1, X_2, \ldots, X_n)$ consist of $n$ i.i.d. random variables belonging to $\mathcal{N}(0, \sigma^2)$, and $Z = f(X)$ be a $\gamma$-Lipschitz function, i.e., $|f(X) - f(X')| \leq \gamma\|X - X'\|$. Then, we have,*

$$\Pr(Z - \mathbb{E}[Z] \geq \epsilon) \leq \exp\left(-\frac{\epsilon^2}{2\gamma^2\sigma^2}\right) \text{ for all } \epsilon > 0. \qquad (4)$$

Let $f(X) = Z := \mathbb{E}[e^{\theta\ell(M(X_i))} \mid X_i = x] - \frac{1}{k}\sum_{i=1}^k e^{\theta\ell(m(X_i))}$. Then we show that $f$ is $\frac{\gamma_e}{\sqrt{k}}$-Lipschitz continuous.

**Lemma 4.** *Let $\ell(M(x))$ be $\gamma$-Lipschitz and $\ell(M(x)) \in [0, 1]$ for all $x \in \mathbb{R}^n$. Then, the function $e^{\theta\ell(M(x))}$ is Lipschitz continuous with Lipschitz constant $\gamma_e := \gamma\theta e^\theta$ for bounded $\theta$.*

*Proof.* By the chain rule, we have $\frac{de^{\theta\ell(M(x))}}{dx} = \frac{d\ell(M(x))}{dx}\theta e^{\theta M(x)}$. Recall $\ell(M(x))$ is Lipschitz with constant $\gamma$, that is, $\frac{d\ell(M(x))}{dx} \leq \gamma$. Then, due to the bounded nature of $M(x) \in [0, 1]$, we have $\frac{de^{\theta\ell(M(x))}}{dx} \leq \gamma\theta e^\theta$. $\qquad\square$

**Lemma 5.** *$f$ is Lipschitz-continuous.*

$$|f(X) - f(X')| = |\frac{1}{k}\sum_{i=1}^k e^{\theta\ell(m(X_i))} - e^{\theta\ell(m(X_i'))}|$$

$$\leq \frac{1}{k}\sum_{i=1}^k |e^{\theta\ell(m(X_i))} - e^{\theta\ell(m(X_i'))}| \leq \frac{\gamma_e}{k}\sum_{i=1}^k |X_i - X_i'| \leq \frac{\gamma_e}{\sqrt{k}}\|X - X'\|_2$$

*Proof.* The first line follows directly from the definition $f(X) = Z$. The first inequality follows from Triangle inequality. The second inequality follows follows from Lipschitz-continuity of $e^{\theta m(x)}$. The last inequality uses the $l_1$-$l_2$ inequality. □

Using these lemmas and substituting our definition of $Z$ and the Lip-constant of the lemma in the statement of the Gaussian concentration bound concludes the proof.

### A.3 PROPERTIES OF THE PROPOSED MEASURE:

1. **Monotonicity.** $m(x) \leq m'(x) \Rightarrow R_{\theta,k}(x, m(\cdot)) \geq R_{\theta,k}(x, m'(\cdot))$.

   *Proof.* To see this, recall $R_{\theta,k}(x', m) := (1/\theta) \log \frac{1}{k} \sum_{i=1}^{k} (e^{-\theta m(X_i)})$. From the premise $m(x) \leq m'(x)$ and by the positivity of the risk parameter $\theta > 0$, it is immediate that $-\theta m(x) \geq -\theta m'(x)$. Then, from the monotonicity of the exponential function, it follows that $e^{-\theta m(x)} \geq e^{-\theta m'(x)}$. Then, by the positivity of the risk parameter $\theta$, we have $R_{\theta,k}(x, m(\cdot)) \geq R_{\theta,k}(x, m'(\cdot))$ □

2. **Translation invariance.** For constant $\alpha \in \mathbb{R}$, $R_{\theta,k}(x, m(\cdot) + \alpha) = R_{\theta,k}(x, m(\cdot)) - \alpha$.

   *Proof.* This follows directly from the property of the exponential function that $e^{m(\cdot)+\alpha} = e^{m(\cdot)} e^{\alpha}$ and the property of the log function that $(1/\theta) \log(e^{\alpha}(1/\theta) \log \frac{1}{k} \sum_{i=1}^{k} (e^{-\theta m(X_i)})) = -\alpha + (1/\theta) \log \frac{1}{k} \sum_{i=1}^{k} (e^{-\theta m(X_i)})$. □

3. **Convexity.** For $0 \leq \alpha \leq 1$,
   $$R_{\theta,k}(x, \alpha m(\cdot) + (1-\alpha)m'(\cdot)) \leq \alpha R_{\theta,k}(x, m(\cdot)) + (1-\alpha)R_{\theta,k}(x, m'(\cdot)).$$

   *Proof.* The convexity follows directly from the convexity of the exponential function $e^{-m}$. □

### A.4 BACKGROUND ON MULTI-OBJECTIVE OPTIMIZATION

Consider a non-linear programming problem with inequality constraints such as:

$$\min_{x'} c(x, x') \qquad \text{subject to:} \quad R(x, x') \leq \tau$$

where $c$ and $R$ are regular enough for the mathematical developments to be valid over the feasible region. It is also assumed that the problem has an optimum. Then the sensitivities of the objective function with respect to the threshold $\tau$ can be calculated using the following theorem:

**Theorem 3.** *Castillo et al. (2008) Assume that the solution of the above optimization problem is a regular point and that no degenerate inequality constraints exist. Then, the sensitivity of the objective function with respect to the parameter a is given by the gradient of the Lagrangian function*

$$L = c(x, x') + \lambda^T (R(x, x') - \tau)$$

*with respect to $\tau$ evaluated at the optimal solution $x^*$, i.e.,*

$$\frac{\partial c(x, x^*)}{\partial \tau} = \nabla_\tau L = -\lambda^*$$

where $\lambda^*$ is the dual optimal solution. This shows how much the objective function value $c$ changes when parameter $\tau$ changes.

## B EXPERIMENTS.

### B.1 DATASETS

**HELOC.** The FICO HELOC (FICO, 2018) dataset contains anonymized information about home equity line of credit applications made by homeowners in the US, with a binary response indicating whether or not the applicant has ever been more than 90 days delinquent for a payment. It can be

used to train a machine learning model to predict whether the homeowner qualifies for a line of credit or not. The dataset consists of 10459 rows and 40 features, which we have normalized to be between zero and one.

**German Credit.** The German Credit Dataset (Dua & Graff, 2017) comprises of 1000 entries, each representing an individual who has taken a credit from a bank. These entries are characterized by 20 categorical features, which are used to classify each person as a good or bad credit risk. To prepare the dataset for analysis, we one-hot encoded the data and normalized it such that all features fall between 0 and 1. Additionally, we partitioned the dataset into a training set and a test set, with a 70:30 ratio respectively.

**CTG.** The CTG dataset (Dua & Graff, 2017) consists of 2126 fetal cardiotocograms, which have been evaluated and categorized by experienced obstetricians into three categories: healthy, suspect, and pathological. We process this dataset based on Black et al. (2021). The problem was transformed into a binary classification task, where healthy fetuses are distinguished from the other two categories. We divided the dataset into a training set of 1,700 instances and a validation set of 425 instances. Each instance is described by 21 features, which we normalized to have values between zero and one.

## B.2 Model Architecture

We initially trained a base neural network model. We use this model to generate counterfactuals. The architecture of our base model consists of two hidden layers, each comprising 128 hidden units. For activation, we used the Rectified Linear Unit (ReLU) function and employed the Adam optimizer. The training process involved 50 epochs with a batch size of 32. This model architecture and training setup for three datasets: HELOC, German Credit, and CTG, since it yielded a satisfactory level of accuracy on all of them. To assess the robustness of the counterfactual examples, we proceeded to train 50 additional models, denoted as $M_{new}$. Under various model change scenarios, we evaluated the validity of the counterfactuals using these new models. All 50 models followed the same architecture and training setup as the base model. These modifications encompassed Weight Initialization (WI), involving retraining the models using the same hyperparameters but different weight initialization methods. Specifically, we used distinct random seeds for each model to vary the weight initialization.

## B.3 Results.

Here, we present our experimental results on existing datasets. Our experimental results are consistent with our theoretical analysis. We observe the minimum cost counterfactual may not be robust. The averages are reported alongside with the standard deviations (average $\pm$ standard deviation) reported in the tables. Robust counterfactuals have higher validity at the cost of higher costs. We report LOF, but none of the models incorporate LOF into their algorithm. We report the results for hyperparameters $\tau = 0.8$ and $\sigma = 1$. The risk parameter $\theta$ is specific to our algorithm and is set to $\theta = 1$.

**Remark.** *Our proposed method, Entropic T-Rex, achieves comparable results to the SNS algorithm (Black et al., 2021). The advantage of our algorithm is in the theoretical results that connect it to the robust worst-case models such as ROAR algorithm (Upadhyay et al., 2021).*

Table 2: Experimental results for HELOC dataset.

| **HELOC** | $l_1$ *based* | | $l_2$ *based* | |
|---|---|---|---|---|
| | COST | VAL. | COST | VAL. |
| Closest Counterfactual | 0.45±0.66 | 69% ± 0.16 | 0.14±0.08 | 54%±0.22 |
| **Entropic T-Rex** (ours) | 2.82±0.65 | 100%±0.01 | 0.76±0.11 | 100%±0.02 |
| SNS | 1.25±0.64 | 98%±0.03 | 0.33±0.08 | 99%±0.02 |
| ROAR | 3.14± 0.48 | 100% ± 0.00 | 1.22± 0.20 | 100%± 0.00 |

Table 3: Experimental results for German Credit dataset.

| German Credit | $l_1$ based | | $l_2$ based | |
|---|---|---|---|---|
| | COST | VAL. | COST | VAL. |
| Closest Counterfactual | $1.45 \pm 1.50$ | $60\%\pm5$ | $0.48\pm0.31$ | $27\%\pm 6$ |
| **Entropic T-Rex** (ours) | $4.62\pm1.44$ | $98\%\pm0$ | $1.21\pm0.30$ | $100\%\pm6$ |
| SNS | $2.38\pm1.52$ | $81\%\pm4$ | $0.68\pm0.31$ | $63\%\pm8$ |
| ROAR | $2.06\pm0.39$ | $100\%\pm 0$ | $2.34\pm0.43$ | $100\%\pm0.0$ |

Table 4: Experimental results for CTG dataset.

| CTG | $l_1$ based | | $l_2$ based | |
|---|---|---|---|---|
| | COST | VAL. | COST | VAL. |
| Closest Counterfactual | $0.24\pm0.19$ | $97\%\pm5$ | $0.10\pm0.04$ | $87\%\pm20$ |
| **Entropic T-Rex** (ours) | $2.80\pm0.15$ | $100\%\pm0$ | $0.98\pm0.04$ | $100\%\pm0.0$ |
| SNS | $0.90\pm0.20$ | $100\%\pm0$ | $0.9\pm0.12$ | $100\%\pm0.0$ |
| ROAR | $1.98\pm0.13$ | $100\% \pm 0.0$ | $1.50\pm 0.10$ | $100\%\pm0.0$ |

## C  LIMITATIONS AND IMPLICATIONS.

The integration of Machine Learning (ML) systems into our daily lives has wide-ranging and complex implications. These implications range from economic to societal to ethical and legal considerations, necessitating a comprehensive approach to address the sociotechnical evolution driven by ML. While our current robust counterfactual approach represents a step towards developing trustworthy ML algorithms, it falls short in considering other important factors. Besides the robustness of counterfactuals which was the focal point of our research in this work, counterfactual explanations suffer from a multitude of limitations such as privacy due to leaked model parameters and fairness ((Sharma et al., 2019; Ley et al., 2022)). For example, neglecting fairness in counterfactual generation can result in algorithms that make decisions with significant moral, ethical, social, and legal implications. Consider this scenario, when examining a loan approval, a counterfactual suggesting an increase in the value of the applicant's collateral might be perceived as fairer for an elderly applicant, as opposed to a counterfactual suggesting an increase in education level or student status. Therefore, in our future work, we will explore approaches that incorporate additional metrics beyond explainability and robustness to generate counterfactuals, addressing fairness and other relevant considerations.

The implications of our research on the robustness of counterfactuals extend beyond end-users to practitioners. By ensuring the reliability and trustworthiness of counterfactuals from both user and institutional perspectives, we can foster greater trust in ML systems, leading to broader economic benefits. However, it is important to recognize that achieving the robustness of counterfactuals requires solving computationally more expensive constrained optimization problems compared to unconstrained optimization of the closest counterfactual. Therefore, future efforts should focus on devising efficient algorithms and computational techniques to overcome this challenge and ensure the sustainability of robust counterfactual approaches.

An inherent limitations of robust counterfactual methods is the class of model changes we considered. Our method focuses on the robustness of a classification model with respect to model changes. Though our method is practically implementable and potentially applicable to all model changes and datasets, the probabilistic guarantees in Theorem 2 may not apply to all models or datasets due to our assumptions. A limitation of our work is the dependence of our probabilistic guarantee on the Lipschitz constant of the models since a large Lipschitz constant can weaken the robustness guarantees. The reliance of our method on sampling and gradient estimation has the drawback of having high computational complexity.

## D    RELATED WORK.

The importance of robustness in local explanation methods was studied in works as early as (Hancox-Li, 2020), but only a handful of recent works have explored algorithms for generating robust counterfactuals. Existing literature has looked at both worst-case robust approaches, e.g. Upadhyay et al. (2021), as well as robustness-constrained approaches, e.g. Black et al. (2021) and Hamman et al. (2023). The robustness-constrained methods quantify counterfactual robustness within a Lipschitz model's local neighborhood, integrating it as a constraint in optimization to ensure the explanation's validity under model variations. In contrast, the worst-case robust techniques (Upadhyay et al., 2021) hedge against the worst model within a model class, employing min-max optimization to find robust counterfactuals. Notably, the connection between these two approaches has not been studied in existing literature. Other related works on the robustness of counterfactuals to model changes include (Rawal et al., 2020; Dutta et al., 2022; Jiang et al., 2022; Nguyen et al., 2022). Upadhyay et al. (2021) propose ROAR that uses min-max optimization (a worst-case approach) to find robust counterfactuals. Rawal et al. (2020) focus on analytical trade-offs between validity and cost. Jiang et al. (2022) introduce a method for identifying close and robust counterfactuals. Black et al. (2021) propose the Stable Neighbor Search (SNS) algorithm that uses local Lipschitzness to generate robust (consistent) counterfactuals for neural networks. Hamman et al. (2023) propose TRex that uses Gaussian sampling around the counterfactual to provide robust counterfactuals. Dutta et al. (2022) focus only on tree-based models (non-differentiable). Mishra et al. (2021) provide a more detailed survey of the literature on counterfactual explanations prior to 2021. We also note that there have been alternative perspectives on the robustness of counterfactuals. Two notable examples include the work by (Laugel et al., 2019; Alvarez-Melis & Jaakkola, 2018) proposes an alternative perspective of robustness in explanations (called L-stability in Alvarez-Melis & Jaakkola (2018)) which is built on similar individuals receiving similar explanations, and the work by (Pawelczyk et al., 2022; Maragno et al., 2023; Dominguez-Olmedo et al., 2022) that focus on robustness to input perturbations (noisy counterfactuals) rather than model changes.

In contrast to these and similar to (Upadhyay et al., 2021; Rawal et al., 2020; Black et al., 2021; Dutta et al., 2022; Jiang et al., 2022; Nguyen et al., 2022; Hamman et al., 2023) , our focus is on counterfactuals remaining valid after model changes.

Entropic risk measure has been the cornerstone of risk-sensitive control (see Jacobson (1973); Speyer et al. (1974); Kumar & Van Schuppen (1981); James et al. (1994); Baras &  (1997); James & Baras (1995); James & Baras (1996); Baras & Patel (1998)) and risk-sensitive Markov decision processes (see Howard & Matheson (1972)). In the context of controls, the connection between risk-sensitive control and robust control has been shown in its full generality (James et al., 1994), establishing that the entropic risk measure emerges from the mathematical analysis of H-infinity output robust control for general non-linear systems. Further analytical development of such mathematical analysis for financial applications has been studied extensively; see  Föllmer & Schied (2002) and references therein.

