# OpenReview forum: "An Entropic Risk Measure for Robust Counterfactual Explanations"
_ICLR.cc/2024/Conference — Submitted to ICLR 2024_

### Official Review · Reviewer_S55q · 2023-10-25

**Soundness:** 3 good
**Presentation:** 2 fair
**Contribution:** 2 fair
**Rating:** 5
**Confidence:** 4

**Summary:**

The paper proposes a methodology for computing counterfactual explanations that are robust to model changes.

**Strengths:**

- Relevant research question
- Novel and promising approach based on entropic-risk

**Weaknesses:**

- Describe exiting methods like SNS and ROAR in more detail
- Only Neural networks are considered in the experiments. What about other classes of models? => ROAR method seems to be strong competitor!
- Lipschitz Countinuity is a very strong assumption -- the authors acknowledge this though
- Readability and understandability could be improved by describing the final algorithm in more detail and in a more structured way (e.g. using pseudo-code). Right now it is only described in textual form in the paragraph "Algorithmic Strategy"

While reading the paper I had a few questions which got partially answered in the end. Maybe those could be addressed earlier in the paper:
- What about the computational complexity? How does it compare to the other methods? (this is only briefly mentioned in the end)
- Are true gradients always needed? This assumes access to the model. What about black-box models? (also only briefly mentioned in the end)

**Questions:**

- Plausibility and actionability are also very important aspects in recourse -- how could these be added to the proposed method?

---

> ### Author Response · Authors · 2023-11-19
>
> We thank the reviewer for their positive evaluation of our work and their comments and suggestions. We also thank the reviewer for acknowledging that our research problem is “relevant” and our approach is “novel and promising”.
>
> **(1 of 3) Regarding the gradient and computation complexity:** The algorithm does not use the actual gradients but only the estimates. Our approach shares comparable computational complexity with T-Rex. As discussed in Section 5, it is important to note that the reliance of our method on sampling and gradient estimation introduces a downside, namely, an elevated level of computational complexity.
>
> **(2 of 3) Regarding the inclusion of additional metrics:** We appreciate the reviewer's insight regarding the inclusion of additional metrics like actionability, plausibility, and fairness. This consideration is acknowledged in the final sentence of the first paragraph of Appendix C, where we recognize the importance of addressing these factors beyond explainability and robustness. Dealing with multi-criteria optimization involves techniques from multi-objective optimization, and we plan to delve into the necessary trade-off analysis using multi-criteria optimization in future research. Thank you for sharing this with us. We make sure to explain these thoroughly in an earlier part of the paper.
>
> **(3 of 3) Regarding readability and understandability**: We acknowledge the importance of a readable and understandable manuscript. We are committed to incorporating your suggestions.

---

### Official Review · Reviewer_wXka · 2023-10-29

**Soundness:** 2 fair
**Presentation:** 2 fair
**Contribution:** 2 fair
**Rating:** 3
**Confidence:** 4

**Summary:**

In this paper, the authors focus on the robustness of counterfactual explanations to potential model changes rather than the data perturbation or contamination. Instead of using the worst possible model to mitigate the issue in the existing literature, they adopt the idea of randomization of a certain model by introducing a prior probability distribution P over the set of possible models. They then
consider the application of the so-called entropic risk measure, as a quantification of robustness for counterfactuals, to hedge against the model’s uncertainty based on their probability of occurrence.

Due to the extreme value property of the entropic risk measure (with respect to the risk aversion parameter θ), they build the connection between their model and the existing worst-case robust model, i.e., model (P2), where the latter can be seen as the limit case of the former when θ Ñ 8. When the prior probability distribution of model uncertainty P is unknown or complex, they propose to use the sampling of the input data around the counterfactual to estimate the entropic risk measure of the model uncertainty. Under some condition that the MGFs of the model uncertainty at the counterfactuals and the output of the original model at points chosen randomly around the counterfactuals are sufficiently close, they derive the so-called “finite sample guarantee” of the estimation of
entropic risk measure for the model uncertainty. Finally, some numerical results are provided.

**Strengths:**

First, they propose entropic risk as a novel measure of robustness for counterfactuals, which establishes a connection between the worst-case robust and robustness-constrained counterfactuals. The new approach covers the worst-case scenario when the risk parameter takes extreme value. Second, they propose a relaxed entropic risk measure which is computable when the distribution of the output of the changed model at the counterfactual point is unknown. Third, the probabilistic guarantees are derived for the proposed robustness metric under a class of model changed based on their moment-generating function.

**Weaknesses:**

I feel that the idea of relating robust optimization to convex risk measures is not new. Thus, from the technical or theoretical aspect, the manuscript is not novel.

The main limitations of the paper are threefold. First, there is the challenge of obtaining the prior probability distribution (P) of model changes. Second, it is difficult to validate the MGF $(\tilde{\epsilon}, \theta)$-equivalence condition as it is too complicated in general. The last limitation concerns the reliability of the counterfactual for the estimated model (P5).

**Questions:**

What motivated the choice of the entropic risk measure over other commonly used risk measures, such as conditional value at risk, as the latter is easy to estimate and compute? Could you elaborate on the advantages and specific scenarios where this choice proves most effective? How about the reliability result of the counterfactual for the estimated model?

When the distribution of the changed model M is unknown, an alternative approach is to construct an ambiguity set, such as a ball centered at an empirical probability under some distance (e.g. Wasserstein metric), and consider a DRO model which may have a good
out-of-sample performance. Can you give some comments on this

---

> ### Author Response · Authors · 2023-11-19
>
> We thank the reviewer for their comments and questions.
>
> **(1 of 3) Regarding our assumptions**: We have not assumed a known probability distribution. We recognize the inherent challenge in verifying the MGF-equivalence assumption. As acknowledged on page 7, we intentionally choose relatively strong assumptions on MGFs within this framework, primarily for illustrative purposes to showcase the potential for algorithmic advancements based on our theoretical insights. Relaxing this assumption, though demanding more intricate proofs, paves the way for less assertive yet still valuable probabilistic guarantees.
>
> **(2 of 3) Regarding the choice of the risk measure and its implications**: Entropic risk measure enables the connection to worst-case robust approaches and offers a unifying perspective. Also, it is known the entropic risk measure works with the tail of the distribution.
> Analogous to the significance of **Renyi mutual information over mutual information** [Mironov (2017); Baharlouei (2020)] that arises in other domains such as fairness, privacy, communications, etc. Reyni measures introduce a tunable $\alpha$ parameter to bridge between worst case and average case. In the limit of $\alpha$ going to infinity, the measure converges to the worst-case expression (maximum), while in the limit of $\alpha$ going to 1, the measure converges to an average-case expression (mutual information).
>
> Similarly, the significance of our contribution lies in the insights that our measure offers for the theoretical understanding of robust counterfactuals, along with the novel **unification** it brings to various approaches within this domain. Our metric has a ‘knob’, the risk parameter $\theta$, that can be adjusted to trade-off between risk-constrained and worst-case/adversarially robust approaches, depending on where the user wants to be. **While risk-constrained accounts for general model changes in an expected sense, the adversarial robustness prioritizes the worst-case perturbations to the model, thus having a higher cost.** **Our approach enables one to tune “how much” a user wants to prioritize for the worst-case model changes, by trading off cost.** The extreme value of the knob (risk parameter $\theta$) maps our measure back to a min-max/adversarial approach.
>
> Notably, (i) Without such a tunable knob, it is not possible to have the flexibility to tradeoff between Risk-Constrained and Worst-Case. **To the best of our knowledge, existing Risk-Constrained measures, e.g., T-Rex, SNS, etc. do not have such an analogous tunable parameter that enables them to mathematically converge to Adversarial (Worst-Case) Robustness.** To clarify, they do have other hyperparameters but varying them would not lead to the mathematical equivalence to **Adversarial (Worst-Case) Robustness**. (ii) It also enables us to characterize the pareto-optimal front and balance cost and risk by varying the tunable risk parameter $\theta$ which is not present in Hamman et. al. [2023].
>
> [Mironov (2017)] Mironov, Ilya. "Rényi differential privacy." 2017 IEEE 30th computer security foundations symposium (CSF). IEEE, 2017;
>
> [Baharlouei(2019)]  Sina Baharlouei, Maher Nouiehed, Ahmad Beirami, Meisam Razaviyayn: Rényi Fair Inference. ICLR 2020
>
> **(3 of 3) Regarding the distributionally robust optimization**: The literature recognizes the equivalence between risk-optimization and distributionally robust optimization; for instance, the entropic risk measure is equivalent to a distributionally robust formulation with a KL-divergence ball as the ambiguity set. Different risk measures correspond to distinct ambiguity sets. Notably, risk measures prove more amenable to gradient descent-based techniques compared to distributionally robust optimization methods.

---

### Official Review · Reviewer_RVaH · 2023-10-30

**Soundness:** 2 fair
**Presentation:** 2 fair
**Contribution:** 1 poor
**Rating:** 3
**Confidence:** 4

**Summary:**

The authors study counterfactual explanations which are robust against changes in the prediction model parameters. To this end they propose an optimization problem where the costs of changing a factual instance x to a counterfactual instance x' is minimized while the risk of remaining a CE after model change is constrained to be smaller than a user-chosen parameter tau. The authors derive theoretical results that their proposed risk measure can be adjusted by the risk parameter to navigate between the robustness constrained methods and worst-case robust methods. Furthermore they provide a relaxation based on point samples around the CE which can be applied e.g. when the underlying distribution is not known. Finally the method is compared to other state-of-the-art methods on several datasets.

**Strengths:**

The topic of the paper is highly relevant and well-studied in the literature. The authors replace the maximum expression used in the worst-case approach by a risk-measure which is a new and innovative idea. The paper is clearly written and the results are proved in detail and are not trivial. The property of controlling the risk of the risk measure by the parameter theta is a nice feature.

**Weaknesses:**

In my opinion the paper has several weaknesses. First, there is no argumentation why the chosen risk measure is useful and why it should be chosen above other risk measures or already used risk measures. Furthermore, the paper is not able to convince the reader that using this risk measure is an improvement compared to the state-of-the-art methods. While the approach provides a nice theoretical connection to the worst-case approach the experiments actually show that the SNS method leads to a much better trade-off between costs and robustness. While the SNS method has only a small reduction in robustness the reduction in costs is significant. Furthermore in the experiments there is no comparison with the most related method T-Rex.

Minor Issues:
- p.4 first paragraph: l is defined on M\times X but takes only one input M(x)
- Example 1 and 2 need explanation (or citation)
- p.5. reduced the risk -> increases the risk by the same factor
- p.7, something is missing in the sentence "However, in cases where our setup permits us to assume
that both the distributions of m(X_i) and M(x') characteristics"

**Questions:**

- Why is the chosen risk measure an improvement regarding the state-of-the-art methods?
- How do I choose my risk-parameter theta as a user of your method?
- Why should I use your method while the SNS method provides much better tradeoff between costs and robustness?
- How can the costs of the SNS method (with l2 norm) on the HELOC dataset be smaller than the closes counterfactual point? The latter should have the smallest costs by definition right?

---

> ### Author Response · Authors · 2023-11-19
>
> We thank the reviewer for their feedback and for acknowledging that the paper is “clearly written”, the problem is “relevant,” our idea is “innovative,” and our results are “detailed” and “non-trivial”.
>
> **(1 of 5) Regarding the choice of risk measure**: Entropic risk measure enables a theoretical connection between our method (entropic-risk-based approach) and the worst-case robust counterfactuals (min-max optimization), e.g. ROAR, unifying these two classes of robust counterfactual methods \textbf{under model changes}. The novelty of our measure is analogous to the significance of **Renyi mutual information over mutual information** [Mironov (2017); Baharlouei (2020)] that arises in other domains such as fairness, privacy, communications, etc. Reyni measures introduce a tunable $\alpha$ parameter to bridge between worst case and average case. In the limit of $\alpha$ going to infinity, the measure converges to the worst-case expression (maximum), while in the limit of $\alpha$ going to 1, the measure converges to an average-case expression (mutual information).
>
> Similarly, the significance of our contribution lies in the insights that our measure offers for the theoretical understanding of robust counterfactuals, along with the novel **unification** it brings to various approaches within this domain. Our metric has a ‘knob’, the risk parameter $\theta$, that can be adjusted to trade-off between risk-constrained and worst-case/adversarially robust approaches, depending on where the user wants to be. **While risk-constrained accounts for general model changes in an expected sense, the adversarial robustness prioritizes the worst-case perturbations to the model, thus having a higher cost.** **Our approach enables one to tune “how much” a user wants to prioritize for the worst-case model changes, by trading off cost.** The extreme value of the knob (risk parameter $\theta$) maps our measure back to a min-max/adversarial approach.
> [Mironov (2017)] Mironov, Ilya. "Rényi differential privacy." 2017 IEEE 30th computer security foundations symposium (CSF). IEEE, 2017;
>
> [Baharlouei(2019)]  Sina Baharlouei, Maher Nouiehed, Ahmad Beirami, Meisam Razaviyayn: Rényi Fair Inference. ICLR 2020
>
>
> Notably, (i) Without such a tunable knob, it is not possible to have the flexibility to tradeoff between Risk-Constrained and Worst-Case. **To the best of our knowledge, existing Risk-Constrained measures, e.g., T-Rex, SNS, etc. do not have such an analogous tunable parameter that enables them to mathematically converge to Adversarial (Worst-Case) Robustness.** To clarify, they do have other hyperparameters but varying them would not lead to the mathematical equivalence to **Adversarial (Worst-Case) Robustness**. (ii) It also enables us to characterize the pareto-optimal front and balance cost and risk by varying the tunable risk parameter $\theta$.
>
>
> **(2 of 5) Regarding the benchmarking**: Our main contribution is to demonstrate a theoretical connection between our method (entropic-risk-based approach) and the worst-case robust counterfactuals (min-max optimization), e.g. ROAR. Our experiments simply aim at showcasing that our method performs at par with the state-of-the-art, SNS algorithm, while being grounded in the solid theoretical foundation of large deviation theory, without relying on any postulations; enabling a unifying theory for robust counterfactuals. We acknowledge the existence of several intriguing techniques in robust counterfactuals that might surpass our strategy in specific scenarios. However, the practical implications of model change remain uncertain. Our fundamental innovation lies in making a theoretical contribution that facilitates a nuanced trade-off with robustness.

---

> > ### Author Response · Authors · 2023-11-19
> >
> > **(3 of 5) Regarding T-Rex**: Our algorithm can be primarily described as substituting our metric for the (relaxed) Stability function of the algorithm presented in Hamman et al. [2023] and indeed we even call our implementation Entropic T-Rex. We have acknowledged this in section 4 (subsection 4.1). Our algorithm also enables us to characterize the Pareto-Optimal front and balance cost and risk by varying the tunable risk parameter $\theta$ which is not present in Hamman et. al. [2023]. However, the true value of our contribution lies in the insights this measure offers for the theoretical understanding of robust counterfactuals, along with the unification it brings to various methods within this domain. Below, we outline some of the key distinctions:
> >
> > **(i) From Risk-Awareness to Robustness**: Our metric has a ‘knob’, the risk parameter $\theta$, that can be adjusted to trade-off risk (robustness) and cost of a counterfactual. This enables: **(i-a) A Trade-off Analysis**: Characterizing the pareto-optimal front and therefore balancing cost and risk using the parameters of our metric. Such analysis is not present in Hamman et. al. [2023].  **(i-b) Theoretical connection to min-max**: The extreme value of the risk parameter maps our measure back to a min-max approach. Hamman et. al. (and for this matter SNS) postulate a risk measure and show empirical "robustness" with respect to model changes. However, there is no mathematical proof that using this measure leads to a robust (worst-case) method. Our method shows the robust properties of our algorithm by connecting this method to the min-max approaches.
> >
> > **(ii) Probabilistic guarantees**: There is a subtle but consequential difference in our probabilistic guarantee that is enabled by the choice of our metric. We introduce the entropic risk measure as a measure of robustness for counterfactuals and provide a probabilistic guarantee on how this measure is related to its relaxed (computable) measure employed in our algorithm. In contrast, the guarantees in Hamman et. al. pertains to their (non-computable) risk measure and is not relevant to their computable proxy Stability measure. Thus, their probabilistic guarantees do not apply to their (relaxed) metric and in turn their algorithm.
> >
> > **(4 of 5) Choice of the risk parameter $\theta$**: The optimal value of this parameter is contingent upon various factors, such as the specific data distribution, making it impractical to analytically set for an unknown distribution. However, a useful guideline for tuning this hyperparameter is as follows: Notably, as the parameter approaches zero, we revert to the risk-neutral case. Consequently, one can initiate the tuning process with a value very close to zero and employ a Backtracking line search to incrementally increase the parameter towards a higher value, enhancing robustness, before the problem becomes infeasible.
> >
> > **(5 of 5) Regarding Minor issues**: Thank you for bringing these to our attention. We have promptly corrected them.

---

> > > ### Comment · Reviewer_RVaH · 2023-11-22
> > > **Replying to authors' response**
> > >
> > > Thank you for the response. I read it. The authors mainly repeat what they had already written in the submission. I keep my score as it is.

---

### Official Review · Reviewer_5rKA · 2023-10-30

**Soundness:** 3 good
**Presentation:** 2 fair
**Contribution:** 2 fair
**Rating:** 3
**Confidence:** 4

**Summary:**

This paper studies the problem of robustness in counterfactual explanations under model shifts. The approach introduced in this paper leverages the entropic risk measure to assess the model's robustness. However, the entropic risk measure is not practically computable, as it needs sampling across the space of shifted models. Therefore, the authors derive an upper bound for the entropic risk measure, which can be feasibly computed by sampling the input space around the counterfactual. Subsequently, they generate counterfactuals by using a counterfactual generation algorithm similar to the one presented by Hamman et al. (2023).

**Strengths:**

The paper studies an important problem in counterfactual explanations literature.

**Weaknesses:**

- My primary concern is about the novelty and significance of the proposed method.
- Experiments are not very comprehensive. More recent baselines and datasets are needed.
- The paper is relatively well-written. However, it is not clear about the novelty or significance of the proposed method compared to previous work. The authors have included some experimental details, but they have not provided code for the paper.

**Questions:**

- It is not clear about the connection between the proposed method and Hamman et al. (2023). Can we directly substitute the entropic risk measure with the stability in Algorithm 1 and Algorithm 2 of Hamman et al. (2023)?
- In Theorem 1, I acknowledge that Lemma 1 is a result of previous work. Therefore, Lemma 2 seems to be the main technical contribution. However, its significance remains unclear to me.
- The probabilistic guarantees in Theorem 2 are based on the assumption of Lipschitz continuity, and it might limit the application of this paper.
- Regarding Theorem 2, we want $k$ to be large to keep a high probability bound. Then, how do you choose the number of samples $k$ in practice?
- The proposed method demonstrates superior performance compared to ROAR in the HELOC dataset. However, in the German Credit and CTG datasets, the cost of counterfactual with Entropic T-Rex is significantly higher than ROAR. In my opinion, it would be more beneficial to report the trade-off between cost and validity instead of reporting these metrics separately.
- The authors should compare to more recent baselines such as RBR (Nguyen et al. UAI, 23)
- Some citations should be mentioned:
1. Counterfactual Plans under Distributional Ambiguity, ICLR22
2. Provably Robust and Plausible Counterfactual Explanations for Neural Networks via Robust Optimisation, ACML23
3. Distributionally Robust Recourse Action, ICLR23
4. On Minimizing the Impact of Dataset Shifts on Actionable Explanations, UAI23

---

> ### Author Response · Authors · 2023-11-19
>
> We thank the reviewer for their feedback. We appreciate their acknowledgment of our work being “relatively well-written” and the problem being “important in the literature”. We are committed to further refining it based on their constructive suggestions.
>
> **(1 of 6) Regarding novelty and significance of the proposed method**: The novelty of our measure is analogous to the significance of **Renyi mutual information over mutual information** [Mironov (2017); Baharlouei (2020)] that arises in other domains such as fairness, privacy, communications, etc. Reyni measures introduce a tunable $\alpha$ parameter to bridge between worst case and average case. In the limit of $\alpha$ going to infinity, the measure converges to the worst-case expression (maximum), while in the limit of $\alpha$ going to 1, the measure converges to an average-case expression (mutual information).
>
> Similarly, the significance of our contribution lies in the insights that our measure offers for the theoretical understanding of robust counterfactuals, along with the novel **unification** it brings to various approaches within this domain. Our metric has a ‘knob’, the risk parameter $\theta$, that can be adjusted to trade-off between risk-constrained and worst-case/adversarially robust approaches, depending on where the user wants to be. **While risk-constrained accounts for general model changes in an expected sense, the adversarial robustness prioritizes the worst-case perturbations to the model, thus having a higher cost.** **Our approach enables one to tune “how much” a user wants to prioritize for the worst-case model changes, by trading off cost.** The extreme value of the knob (risk parameter $\theta$) maps our measure back to a min-max/adversarial approach.
>
> Notably, (i) Without such a tunable knob, it is not possible to have the flexibility to tradeoff between Risk-Constrained and Worst-Case. **To the best of our knowledge, existing Risk-Constrained measures, e.g., T-Rex, SNS, etc. do not have such an analogous tunable parameter that enables them to mathematically converge to Adversarial (Worst-Case) Robustness.** To clarify, they do have other hyperparameters but varying them would not lead to the mathematical equivalence to **Adversarial (Worst-Case) Robustness**. (ii) It also enables us to characterize the Pareto-Optimal front and balance cost and risk by varying the tunable risk parameter $\theta$ which is not present in T-Rex-like algorithms.
>
> [Mironov (2017)] Mironov, Ilya. "Rényi differential privacy." 2017 IEEE 30th Computer Security Foundations Symposium (CSF). IEEE, 2017;
>
> [Baharlouei(2019)]  Sina Baharlouei, Maher Nouiehed, Ahmad Beirami, Meisam Razaviyayn: Rényi Fair Inference. ICLR 2020
>
> **(2 of 6) Regarding the experiments**: Our main contribution is to demonstrate a theoretical connection between our method (entropic-risk-based approach) and the worst-case robust counterfactuals (min-max optimization), e.g. ROAR. Our experiments simply aim at showcasing that our method performs at par with the state-of-the-art while being grounded in the solid theoretical foundation of large deviation theory, without relying on any postulations; enabling a unifying theory for robust counterfactuals. We acknowledge the existence of several intriguing techniques in robust counterfactuals that might surpass our strategy in specific scenarios. However, the practical implications of model change remain uncertain. Our fundamental innovation lies in making a theoretical contribution that facilitates a nuanced trade-off with robustness.

---

> > ### Author Response · Authors · 2023-11-19
> >
> > **(3 of 6) Regarding our algorithm and T-Rex**:
> > Our algorithm can be primarily described as substituting our metric for the (relaxed) Stability function of the algorithm presented in Hamman et al. [2023] and indeed we even call our implementation Entropic T-Rex. We have acknowledged this in section 4 (subsection 4.1). Our algorithm also enables us to characterize the Pareto-Optimal front and balance cost and risk by varying the tunable risk parameter $\theta$ which is not present in Hamman et. al. [2023]. However, the true value of our contribution lies in the insights this measure offers for the theoretical understanding of robust counterfactuals, along with the unification it brings to various methods within this domain. Below, we outline some of the key distinctions:
> >
> > **(i) From Risk-Awareness to Robustness**: Our metric has a ‘knob’, the risk parameter $\theta$, that can be adjusted to trade-off risk (robustness) and cost of a counterfactual. This enables: **(i-a) A Trade-off Analysis**: Characterizing the pareto-optimal front and therefore balancing cost and risk using the parameters of our metric. Such analysis is not present in Hamman et. al. [2023]. **(i-b) Theoretical connection to min-max**: The extreme value of the risk parameter maps our measure back to a min-max approach. Hamman et. al. (and for this matter SNS) postulate a risk measure and show empirical "robustness" with respect to model changes. However, there is no mathematical proof that using this measure leads to a robust (worst-case) method. Our method shows the robust properties of our algorithm by connecting this method to the min-max approaches.
> >
> > **(ii) Probabilistic guarantees**: There is a subtle but consequential difference in our probabilistic guarantee that is enabled by the choice of our metric. We introduce the entropic risk measure as a measure of robustness for counterfactuals and provide a probabilistic guarantee on how this measure is related to its relaxed (computable) measure employed in our algorithm. In contrast, the guarantees in Hamman et. al. pertain to their (non-computable) risk measure and is not relevant to their computable proxy Stability measure. Thus, their probabilistic guarantees do not apply to their (relaxed) metric and in turn their algorithm.
> >
> >
> > **(4 of 6) Regarding Theorem 1**: The significance of Theorem 1 lies in its utilization of Vardhan's theorem (Lemma 1) in conjunction with Lemma 2 in this specific setup. While robust counterfactuals have been studied, Theorem 1 enables us to bridge the gap between the two existing classes of algorithms in this space, namely worst-case min max and average-case risk-constrained. Lemma 2 played a crucial role in bridging the gap between Vardhan's theorem and the main objective of the proof. Lemma 2 shows that supremum over a (compact set) probability simplex is equal to the maximum. This might be in agreement with intuition but not trivial.
> >
> > **(5 of 6) Regarding Theorem 2**: We agree with the reviewer and have acknowledged the dependence on the Lipschidtz constant and its implications in the limitation section in the appendix.
> >
> > **(6 of 6) Regarding references**: Thank you for pointing out the references. We included a brief note and discussed these references in our paper.

---

### Meta-Review · Area_Chair_DSHk · 2023-12-10

**Metareview:**

The paper focuses on the robustness of counterfactual explanations to changes in the prediction model they aim to explain. They do so by relying on an entropic rick measure that allows them to navigate between  constrained methods and worst-case robust methods. However, as pointed by the reviewers, the paper presents major issues with regards to the motivation and practical limitations of the proposed approach, as well as positioning with respect to prior related works. Unfortunately, the major points raised by the reviewers were not convincingly addressed by the authors in the rebuttal period.  I thus encourage the authors to address the detailed feedback provided by the reviewers to further improve their work.

**Justification For Why Not Higher Score:**

Clear rejection case, with an unconvincing rebuttal that did not really address several of the detailed issues raised by the authors.

**Justification For Why Not Lower Score:**

N/A

---

### Decision · Program_Chairs · 2024-01-16

Reject